# Molecular time estimates for the Lagomorpha diversification

**Leandro Iraçabal** [ID][1][☯], **Matheus R. Barbosa**[1][☯], **Alexandre Pedro Selvatti**[2]*, **Claudia Augusta de Moraes Russo** [ID][1]*

1 Departamento de Genética, Rio de Janeiro, Universidade Federal do Rio de Janeiro, CCS, Instituto de Biologia, Rio de Janeiro, Brazil, 2 Departamento de Zoologia, Universidade do Estado do Rio de Janeiro, Instituto de Biologia Roberto Alcântara Gomes, Maracanã, Rio de Janeiro, Brazil

☯ These authors contributed equally to this work.
* apselvatti@gmail.com (APS); claudiaamrusso@gmail.com (CAMR)

**Data Availability Statement:** All relevant data are in public databases or within the manuscript and its Supporting Information files.

## Abstract

Despite their importance as members of the Glires group, lagomorph diversification processes have seldom been studied using molecular data. Notably, only a few phylogenetic studies have included most of the examined lagomorph lineages. Previous studies that included a larger sample of taxa and markers used nonconservative tests to support the branches of their proposed phylogeny. The objective of this study was to test the monophyly of families and genera of lagomorphs and to evaluate the group diversification process. To that end, this work expanded the sampling of markers and taxa in addition to implementing the bootstrap, a more rigorous statistical test to measure branch support; hence, a more robust phylogeny was recovered. Our supermatrix included five mitochondrial genes and 14 nuclear genes for eighty-eight taxa, including three rodent outgroups. Our maximum likelihood tree showed that all tested genera and both families, Leporidae and Ochotonidae, were recovered as monophyletic. In the *Ochotona* genus, the subgenera *Conothoa* and *Pika*, but not *Ochotona*, were recovered as monophyletic. Six calibration points based on fossils were used to construct a time tree. A calibration test was performed (via jackknife) by removing one calibration at a time and estimating divergence times for each set. The diversification of the main groups of lagomorphs indicated that the origin of the order's crown group was dated from the beginning of the Palaeogene. Our diversification time estimates for Lagomorpha were compared with those for the largest mammalian order, i.e., rodent lineages in Muroidea. According to our time-resolved phylogenetic tree, the leporids underwent major radiation by evolving a completely new morphospace—larger bodies and an efficient locomotor system—that enabled them to cover wide foraging areas and outrun predators more easily than rodents and pikas.

## Introduction

Rabbits, hares, and pikas are part of the mammalian order Lagomorpha. These herbivorous mammals have a highly modified jaw morphology that lacks canine teeth and includes

**Funding:** This study was financially supported by the Coordenação de Aperfeiçoamento de Pessoal de Nível Superior – Education Ministry of Brazil (CAPES) in the form of a grant (Finance Code 001) received by LI. This study was also financially supported by the National Research and Technology Council (CNPq) in the form of a grant (310567/2018-1) received by CAMR and a Master's fellowship award received by MRB. This study was also financially supported by Rio de Janeiro State Research Funding Agency (FAPERJ) in the form of grants (E-26/010.001887/2019, SEI-260003/001170/2020, SEI-260003/012995/2021) received by CAMR and a grant (E-26/203.840/2022) (277324) received by APS. The funders had no role in study design, data collection and analysis, decision to publish, or preparation of the manuscript.

**Competing interests:** The authors have declared that no competing interests exist.

gnawing-specialized sharp incisors that grow continuously [1]. These unique features are shared with their sister group, the species-rich large mammalian order Rodentia. Together, rodents and lagomorphs encompass the Glires grandorder, which belongs to the Euarchontoglires clade [2]. Unlike rodents, lagomorphs have one additional small pair of upper incisors that are located behind the larger pair common to all mammals [3].

There are approximately 90 extant species of Lagomorpha split into two living families, distinguishable by the presence of three molar pairs in the upper jaw in Leporidae and only two in Ochotonidae [1, 4]. The monogeneric Ochotonidae includes the living genus *Ochotona*, which is divided into 30 species of hamster-like pikas. They live in Central Asia, Japan and North America and are distinguished from Leporids by their relatively short ears, small hind feet, and lack of a conspicuous tail [5 p. 127]. The Leporidae family includes 10 genera and 60 species of rabbits and hares that are characterized by elongated ears and large hind feet [6].

Historically, the geographical distribution of the lagomorphs included all continents but not Oceania or any other islands. However, some European lagomorphs have been deliberately introduced in Australia, mostly for hunting purposes, as rabbits remind nostalgic European colonists of their faraway home [7]. Despite their importance, only a handful of studies have included global phylogenetic reconstruction and divergence time estimates on the Lagomorphs. Some phylogenetic studies have been conducted, but their analyses have been restricted to specific genera (*Lepus*: [8, 9]; *Ochotona*: [10–13]) or subgenera (*Ochotona*: [14]; *Pika*: [15–17]; *Conothoa*: [18]) and, therefore, have a limited taxonomic scope.

Other studies have included phylogenetic reconstruction with broader sampling but are still restricted to Leporidae [19–21] or Ochotonidae [22]. Averianov's work, for instance, is an analysis of the phylogeny of 28 extant and extinct leporids based on 31 morphological characters. Unfortunately, morphologically based phylogenies are poorly resolved, as expected because of the low number of morphological characteristics used. On the other hand, the work of Matthee and collaborators [20] used 27 species of leporids with five nuclear and two mitochondrial markers. These authors analysed their data using several tree-building methods that yielded relatively high confidence values, particularly for Bayesian inference. For ochotonids, one of the most comprehensive works is that of Yu et al. [22], who used 23 species but included only two mitochondrial markers.

To date, the two major exceptions in terms of the taxonomic scope of lagomorphs are the work on diversification and biogeography [23] and the supertree by Stoner and collaborators [24]. The latter performed a supertree analysis, compiling phylogenies based on morphological and molecular data from 146 lagomorphic articles. Despite the importance of such comprehensive analyses, supertrees come with statistical problems related to the difficulties assessing the reliability of their branches, among other issues (see [25]).

Conversely, in the work of Ge and coworkers [2013], the authors used three mitochondrial markers and Bayesian inference to reconstruct the history of Lagomorph diversification, including a time tree with a detailed analysis of biogeography. However, the time priors (or time calibrations) used in that study were secondary calibrations that were converted into a single point age for Lagomorpha with no confidence interval. Simulated and empirical data have shown that secondary calibrations and the lack of confidence interval in point calibrations leads to uncertainty on divergence time estimates, especially when the node age is shifted from the true age with falsely high precision [26–28].

Divergence times based on multiple fossil calibrations distributed across the internal nodes tend to be more reliable, as this strategy incorporates good rate variation among branches and drastically reduces the errors from single fossil analyses [27, 29–31]. Thus, in the present work, the sampling of markers and species was expanded, totalling five mitochondrial and 14 nuclear genes and 80 ingroup species, to generate a time tree for Lagomorpha. We used multiple fossils

as time priors, including the root and internal nodes of the tree according to best practices [32]; such calibrations were resampled to ascertain their robustness. In addition, the bootstrap test of support relies on ping, a more conservative and reliable statistical support method, was used to measure the support of the internal branches [33]. The objectives of the article are to detail the temporal diversification process of the order Lagomorpha using a more complete dataset, a more robust time tree and more rigorous statistical analyses than did previous studies.

## Methods

The taxonomic reference of this paper was the Higher Taxonomy database [34] (S1 Table). We included the type species for eight genera, namely, *Lepus* (*L. timidus*), *Nesolagus* (*N. netscheri*), *Ochotona* (*O. dauurica*), *Pentalagus* (*P. furnessi*), *Poelagus* (*P. marjorita*), *Pronolagus* (*P. crassicauatus*), *Romerolagus* (*Romerolagus diazi*) and *Sylvilagus* (*S. floridanus*), as well for four *Ochotona* subgenera: *Ochotona* (*Conothoa*) *roylii*, *Ochotona* (*Lagotona*) *pusilla*, *Ochotona* (*Ochotona*) *dauurica* and *Ochotona* (*Pika*) *alpina*.

Our data matrix included all available DNA sequences in GenBank for Lagomorpha, and as outgroups, three Rodentia species were selected, for a total of eighty-eight taxa sampled from the GenBank database [35]. Five mitochondrial genes and 14 nuclear genes were included as markers. All access codes can be found in S2 Table. The alignment of each marker was performed in MAFFT v.7 [36]. The iterative refinement algorithm FFT-NS-i was used for all the markers, except for the ribosomal subunit 12S, for which the Q-INS-i algorithm was selected because it considers the secondary structure of the RNA. The other parameters were set to their defaults in both cases. The concatenation was performed with the AMAS tool [37], and the matrix was composed of 17,938 sites.

For phylogenetic reconstruction and branch support, the IQ-Tree v. 2.1.3 program [38] was used. The alignment was initially divided into 19 partitions, and the ModelFinder algorithm, as implemented in IQ-Tree, was used to calculate the best molecular evolution model for each partition [39]. The algorithm was also used to estimate the best alignment partition scheme, grouping partitions with the same evolutionary models and avoiding problems associated with overparameterization [40]. The criterion used to choose the models and the partition scheme was Bayesian information (BIC), as it is a more conservative estimate [41]. The final scheme consisted of nine partitions with six different models of sequence evolution.

To remove unstable lineages, 1,000 ultrafast bootstrap pseudoreplicates were used as input for RogueNaRok, a software package that identifies rogue taxa [42]. The algorithms used for identification were strict and majority-rule consensus to explore a greater number of potential rogues [33]. The analysis was performed with the parameter "max. dropset size" set at 1 to 4 for the strict consensus and at 1 and 2 for the majority-rule consensus, aiming to allow the grouping of up to four tips in the calculation of the improvement of the bootstrap value. The threshold used to define a lineage as a rogue was 2 for strict consensus and 0.5 for majority rule consensus. Therefore, *Bunolagus monticularis*, *Lepus starcki*, *Sylvilagus andinus*, *Sylvilagus bachmani*, and *Sylvilagus cunicularis* were removed, decreasing the number of species in the final matrix to 80 lagomorphs. Our analysis focused on the rogue-excluded maximum likelihood analysis bootstrap support, but the phylogenetic tree on which the rogues were kept is available in the Supplemental Material (S1 Fig). After rogue removal, the sequences were aligned, producing a final matrix of 17,942 sites, and the phylogeny was inferred using the cited protocols (Fig 1).

The divergence times were estimated using the RelTime method with the MEGA 11 program [43, 44] with fossil-based calibrations (Table 1). A problem commonly associated with

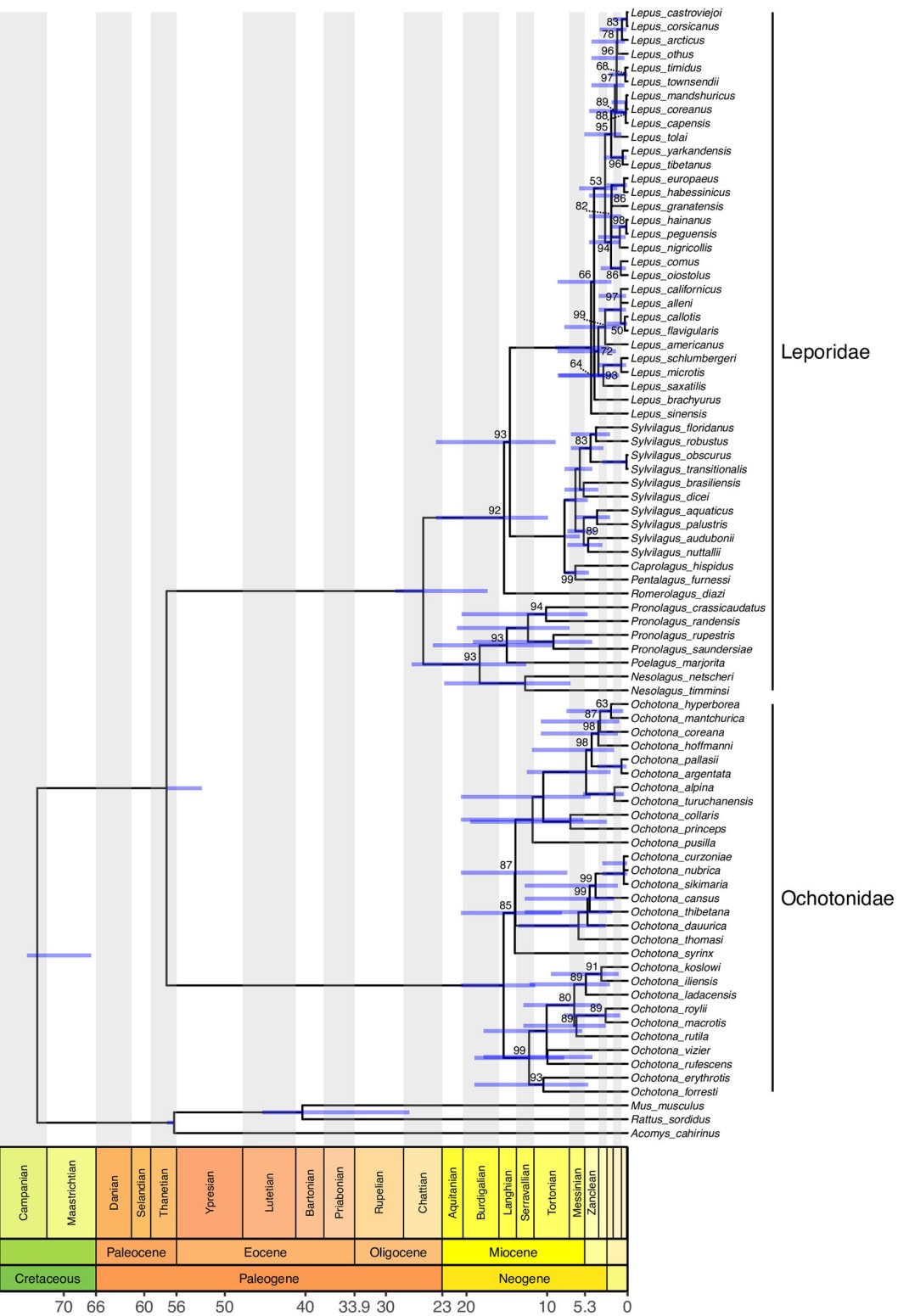

**Fig 1. Phylogenetic relationships and divergence times for Lagomorpha using the maximum likelihood algorithm, 79 terminal taxa, five mitochondrial genes and 14 nuclear genes mitochondrial genes (17,942 sites).** The bootstrap values (1,000 ultrafast bootstrap pseudoreplicates) lower than 100 are shown. All species were considered as Rogue Taxa (Roguenarok program) were removed from the alignment. Timetree produced from a maximum likelihood phylogeny and six calibrations, with 95% confidence intervals shown as bars at nodes. Colors represent different epochs.

**Table 1. Age constraints of six nodes.**

| Calibrated clade | Hard minimum age | Soft maximum age | Lognormal distribution parameters | References |
|---|---|---|---|---|
| Glires | 59.24 | 162.5 | 59.24, 2, 1.35 | Li et al. (2016), Wang et al. (2016), He et al. (2022) |
| Rodentia | 56 | 66 | 56, 1, 0.67 | Marivaux et al. (2004) |
| *Mus-Rattus* | 10.4 | 16 | 10.4, 0.8, 0.47 | Benton et al. (2015) |
| Lagomorpha | 48.07 | 66 | 48.07, 2, 0.45 | Rose et al. (2008) |
| Leporidae | 8 | 11.65 | 8, 0.5, 0.41 | Flynn et al. (2014) |
| Stem *Sylvilagus* | 3 | 4.75 | 3, 0.3, 0.13 | Gazin (1942), Panseko & Lucas (2011) |

non-Bayesian approaches for divergence time inference was once the impossibility of including the calibration data as a probability distribution, but this became possible with the RelTime program [45]. We used a lognormal distribution for all calibrated clades to include uncertainty about the maximum age of the fossils. Initially, seven calibration points previously used to date the lagomorph tree were considered. However, following best practices for the inclusion of fossils in molecular dating [32], only six samples were retained (Table 2). In this case, the calibration of *Lepus* was removed because it was extracted from an indirect estimation [46].

To test the consistency and influence of each calibration point in the analysis, internal consistency tests were performed with several rounds of calibration point sets. In set 1, all six calibration points were included (Table 2), and in the remaining sets, each calibration point was removed in turn. The ages used in the analyses were based on the work of Gradstein et al. [47] on geologic timetables. When a more accurate or up-to-date age delimitation was found in the relevant literature, this was used (see Table 1, references therein).

## Results

In our tree, the Lagomorpha order and the two Lagomorpha families, Leporidae and Ochotonidae, were found to be monophyletic and to have high bootstrap (BP) support values (100% BP for each) (Fig 1). Eight genera were included in our Leporidae clade. Among these, *Lepus* (30 spp.; 100 BP), *Pronolagus* (four species, 100 BP), *Nesolagus* (two species, 100 BP) and *Sylvilagus* (10 species, 100 BP) were recovered as monophyletic clusters, whereas *Caprolagus*, *Pentalagus*, *Poelagus* and *Romerolagus*, the only extant species, were included in our dataset. Among the ochotonids, *Ochotona*, the only extant genus, was also recovered in a tight cluster (29 species, 100 BP).

### Leporidae

In our Leporidae tree (Fig 2), our small Leporidae clade (93 BP), containing the genera *Nesolagus*, *Poelagus* and *Pronolagus*, was found to be the sister of a larger Leporidae clade (45 species,

**Table 2. Time estimates for major Lagomorpha lineages using the seven calibration schemes.**

| | Lagomorpha | Leporidae | Ochotonidae | *Lepus* | *Sylvilagus* | *Nesolagus* | *Pronolagus* |
|---|---|---|---|---|---|---|---|
| Calibration Nodes/Range | 56.00–47.80 | 4.75–3 | 20.44–15.97 | 13.82–11.63 | 5.33–3.60 | - | 7.25–5.33 |
| All included | 57.2 | 25.4 | 15.4 | 4.5 | 6.5 | 12.7 | 12.4 |
| Glires excluded | 62.0 | 27.4 | 16.7 | 4.9 | 6.6 | 13.7 | 13.3 |
| Lagomorpha excluded | 62.5 | 27.1 | 16.8 | 4.7 | 6.1 | 13.6 | 13.2 |
| Leporidae excluded | 57.9 | 26.4 | 15.6 | 4.8 | 7.4 | 13.2 | 12.9 |
| *Mus+Rat* excluded | 57.6 | 25.6 | 15.5 | 4.6 | 6.6 | 12.8 | 12.5 |
| Rodentia excluded | 53.1 | 23.7 | 14.3 | 4.3 | 6.6 | 11.9 | 11.6 |
| *Sylvilagus* excluded | 57.8 | 27.2 | 15.6 | 5.2 | 10.3 | 13.6 | 13.3 |

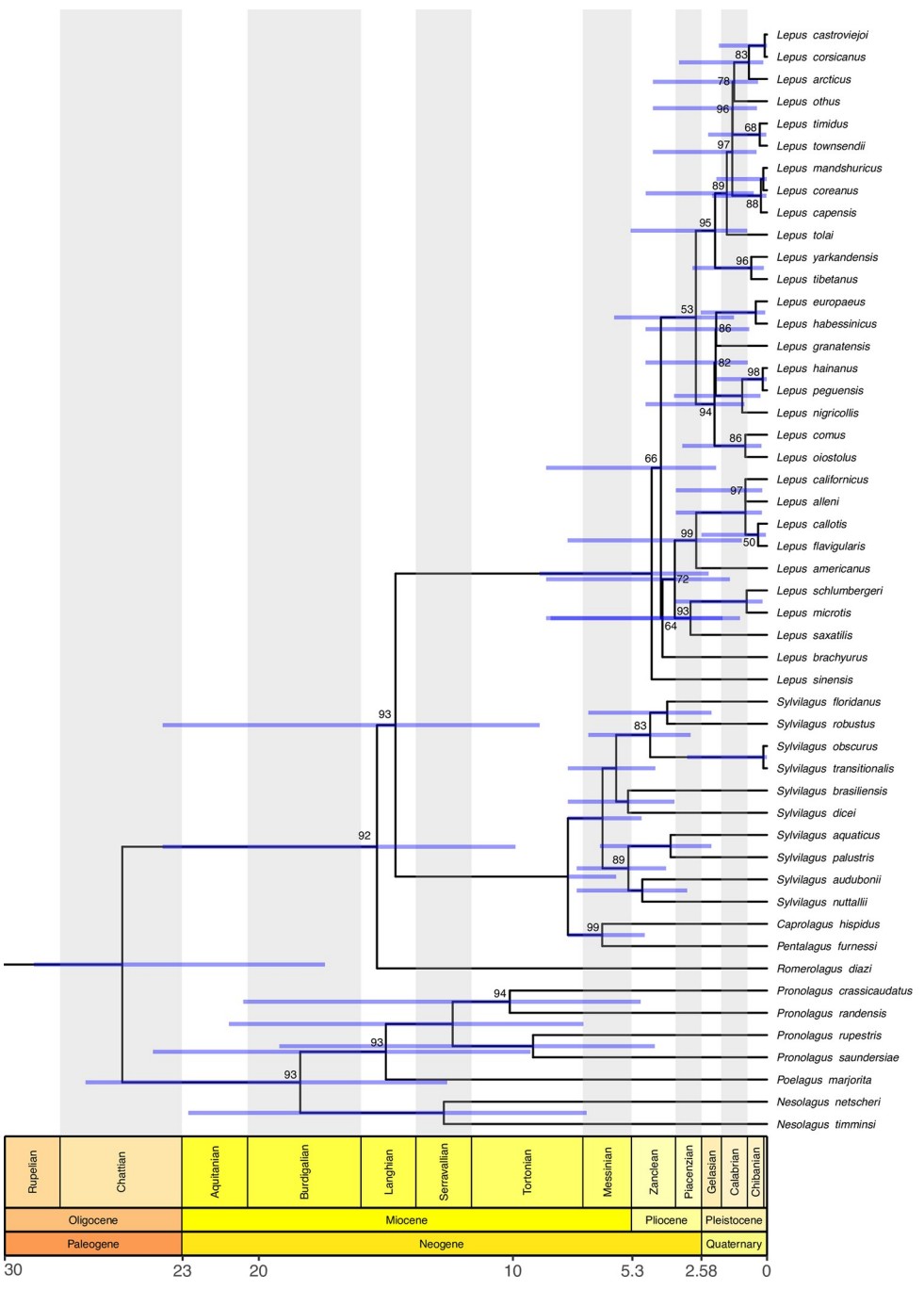

**Fig 2. Phylogenetic relationships and divergence times for Leporidae clade (pruned from Fig 1).**

92 BP) that included the genera *Caprolagus*, *Lepus*, *Pentalagus*, *Romerolagus* and *Sylvilagus* (see also Cano-Sanchéz et al. 2022) (Fig 2). In the small clade, the monotypic *Poelagus* species joined (99 BP) the four-species cluster of *Pronolagus* (4 species 100 BP), in which *P. crassicaudatus* was a sister to *P. randensis* (94 BP) and *P. rupestres* clustered with *P. saundersiae* (100 BP). This lineage was a sister (100 BP) to the *Nesolagus* cluster (100 BP), which included *N. netscheri* plus *N. timminsi*.

In the large Leporidae clade (Fig 2), the monotypic genus *Romerolagus* (*R. diazi*) was a sister to a highly supported group (93 BP) with *Caprolagus*, *Lepus*, *Pentalagus* and *Sylvilagus*. In sequence, the split was between the large genus *Lepus* (30 species, 100 BP) and the diversity that included the *Sylvilagus* clade (10 species, 100 BP) plus the group (2 species, 99 BP) with the monotypic *Pentalagus* and *Caprolagus*.

According to our tree (Fig 2), the *Sylvilagus* cluster is monophyletic (100 BP) and divided into two lineages. The smaller lineage (89 BP) included a two-pair set of species: *S. aquaticus* plus *S. palustris* (100 BP) and *S. audubonii* plus *S. nuttallii* (100 BP). In the larger *Sylvilagus* lineage (100 BP), *S. brasiliensis* joined *S. dicei* (100 BP), and they clustered with another set (83 BP) of two pairs: *S. floridanus* plus *S. robustus* (100 BP) and *S. obscurus* and *S. transitionalis* (100 BP).

Our *Lepus* tree included 30 species (Fig 2), approximately twice as many species as the most comprehensive molecular phylogenetic work to date [23]. In this lineage, the species *Lepus sinensis* was a sister of the remaining *Lepus* diversity (29 species, 66 BP), but support was low. In the main *Lepus* group, the split separates the large clade (21 species, 64 BP), which includes *L. arcticus*, *L. capensis*, *L. castroviejoi*, *L. comus*, *L. coreanus*, *L. corsicanus*, *L. europaeus*, *L. granatensis*, *L. habessinicus*, *L. hainanus*, *L. mandshuricus*, *L. nigricollis*, *L. oiostolus*, *L. othus*, *L. peguensis*, *L. sinensis*, *L. tibetanus*, *L. timidus*, *L. tolai*, *L. townsendii* and *L. yarkandensis* from the smaller clade (nine species, 53 BP) with *L. alleni*, *L. americanus*, *L. brachyurus*, *L. californicus*, *L. callotis*, *L. flavigularis*, *L. microtis*, *L. saxatillis* and *L. schlumbergeri*.

In the small *Lepus* clade (Fig 2), the first split separated *L. brachyurus* from the remaining species into two lineages. In the first lineage (99 BP), *L. americanus* was a sister of the group (100 BP) joining two clades: one is *L. californicus* plus *L. alleni* (97 BP), and the other is *L. callotis* plus *L. flavigularis* (50 BP). In the second (93 BP), *L. schlumergeri* with *L. microtis* (100 BP) was a sister to *L. saxatilis*.

In the large *Lepus* clade, two sister lineages are shown. In the first (53 BP), *L. arcticus* joined (83 BP) *L. castroviejoi* plus *L. corsicanus* (100 BP). This species was a sister to *L. othus* (98 BP), followed by *L. timidus* (68 BP) and *L. townsendii* (97 BP). This group was a sister to *L. capensis* and the *L. mandshuricus* and *L. coreanos* clade (100 BP). *L. tolai* was a sister of this nine-species group (89 BP), and the clade (96 BP) of *L. yarkandensis* and *L. tibetanus* diverged next. This diversity was like that of a group (82 BP) containing two clusters of three species each. In the first (86 BP), *L. granatensis* was a sister of the *L. europaeus* plus *L. habessinicus* clade (100 BP), and in the second (100 BP) clade, *L. nigricollis* was a sister of the *L. hainanus* and *L. peguensis* clade (98 BP, Fig 2).

## Ochotonidae

*Ochotona* traditionally has been divided into four subgenera (*Conothoa*, *Lagotona*, *Ochotona*, and *Pika*) that are distinguished by the type of habitat or geographical region they occupy [11, 48]. A fifth subgenus, *Alienauroa*, was recently proposed based on genomic data [49, 50], and in our study, *Ochotona syrinx* was the sole representative of this subgenus. Lagomorphs of the subgenus *Pika* are ochotonids with a North American distribution, and the subgenus *Ochotona* includes species that live in steppes and shrub environments, whereas, in the subgenus *Conothoa*, we find species that inhabit mountains. The subgenus *Lagotona* was proposed to include a single species, *O. pusilla* [51].

As previously proposed [11], the subgenera *Conothoa* (100 BP) and *Pika* (100 BP) are both monophyletic according to our phylogeny. In our tree, *Ochotona* (*Alienauroa*) *syrinx* was loosely clustered (85 BP) with the clade (87 BP) that included *Ochotona* (*Lagotona*) *pusilla* and the species assigned to the subgenera *Pika* and *Ochotona*. *O. iliensis*, *O. koslowi* and *O. vizier*

were not included in previous subgenus assignments but were tightly grouped (100 BP) in the *Conothoa* subgenus, whereas *O. cansus* and *O. sikimaria* were clustered (100 BP) with high bootstrap support in the *Ochotona* subgenus (Fig 1).

The *Pika* subgenus (100/100) includes North American ochotonids and is arranged as follows in our tree: The first divergence observed is between *O. collaris* + *O. princeps* (100 BP), followed by the divergence of *O. alpina* plus the *O. turuchanensis* clade (100 BP), *O. argentata* plus *O. pallasi* (100 BP), which was the sister of the remaining four species (98 BP), and *O. hoffmanni* as a sister of the polytomy (98 BP), *O. hyperborea* plus *O. mantchurica* plus *O. coreana*.

The shrub and steppe species are the ochotonids of the *Ochotona* clade (100 BP). In the present work, *O. thomasi* was recovered as a sister to the remaining subgenus (100 BP), followed sequentially by *O. dauurica* (99 BP), *O. thibetana* (99 BP), *O. cansus* (99 BP) and *O. sikimaria*, which is the sister of the final *O. curzoniae* + *O. nubrica* (100 BP) clade.

The species that inhabit the mountains make up the *Conothoa* clade (100 BP). In this group, the first split was between the *O. erythrotis* and *O. forresti* clades (93 bp), and the remaining diversity (100 bp) of the subgenus was observed. The next split included the *O. vizier* and *O. rufescens* clades (100 BP) and the remaining *Conothoa* diversity (80 BP). This diversity was divided into two lineages. The first (BP 89) *O. ladecensis* was a sister to *O. koslowi* plus *O. iliensis* (91 BP), whereas in the second (89 BP), *O. rutila* was a sister to *O. roylii* and the *O. macrotis* clade (89 BP).

## Discussion

### Lagomorpha time-resolved phylogenetic tree

The Lagomorphs first differentiated during the Palaeogene, while the Leporidae and Ochotonidae families started to differentiate in the upper Palaeogene and middle Neogene, respectively (Fig 1). The origin of *Lepus* dates from the Pliocene, while the age of *Sylvilagus* dates from the late Miocene. The internal consistency test of the calibrations detected differences between the time estimates when a specific calibration point was absent. However, in most calibration schemes, the ages were congruent. When there was a difference, usually 10% difference, the confidence intervals overlapped. Importantly, the age estimates for Lagomorpha and Leporidae remained stable when their specific calibration points were excluded. However, when the Rodentia calibration was excluded, the age of Lagomorpha recovered to the lower Eocene. Leporidae, on the other hand, originated in the upper Oligocene in every calibration scheme, while Ochotonidae was repeatedly found in the transition between the lower and middle Miocene.

The time tree with all calibration fossils indicated that the beginning of the crown group of lagomorphs was 57.2 million years ago (57.2–52.9 Ma), indicating that it appeared in the Thanetian Age, late Palaeocene, Palaeogene period (Table 2, Fig 1). The estimate by Ge et al. [23], which calibrated only the Lagomorpha node of the ingroup and with a single point in time (50.2 Ma), is younger (50.3 Ma) but consistent with our estimate. Notably, when the calibration for this node was not used in our study, the origin estimate was 62.52 Ma, increasing this difference slightly and with a wider confidence interval (74.8–52.2 Ma). The age proposed by Matthee et al. [20] of 29.0 (±3.8) Ma for Lagomorpha is considerably younger than our age, as these authors calibrated their tree with much younger ages (20–40 Ma) for the nodes. Nevertheless, their calibrations do not seem to correspond to the most recent fossil evidence, which indicates the existence of stem Leporidae and Ochotonidae since the Eocene [51, 52]. In addition, Matthee et al. [20] used a calibration for *Lepus* based on divergence time estimates and not on fossil occurrence. Cano-Sanchéz et al. [53] reported a Palaeocene age of ca. 60 Ma for Lagomorpha.

Among the two extant Lagomorpha clades, Leporidae was the first to diversify; it was dated to 25.4 Ma, the Chattian age, and the upper Oligocene. Ge et al. [23] found an age of approximately 18.1 Ma in the lower Miocene. Halanych and Robinson [54] used the average sequence distance between leporid genera and assumed a molecular clock to calculate the Leporidae divergence time; a slightly older age of approximately 12.2 to 16.3 Ma was found for this family. Similarly, Matthee et al. [20] reported an age of 14 (±1.5) Ma for this node. An even older age of 21.9 (± 3.8) Ma was found by Su and Nei [55] using Ig heavy chain variable region genes. The oldest estimated age for Leporidae was that proposed by Cano-Sanchéz et al. [53], with a divergence occurring ca. 47 Ma in the middle Palaeogene.

Ochotonidae, on the other hand, was recovered at 15.4 Ma in the Langhian, middle Miocene. Melo-Ferreira et al. [16] calibrated only the Lagomorpha node, testing three different ages, 31, 37 and 65 Ma, and found that the ages of Ochotonidae were 6.6, 7.8 and 13.8 Ma, respectively. Notably, no calibration points overlapped our prior distribution for this node (48–59 Ma). When Melo-Ferreira et al. [16] employed the oldest calibration (65 Ma), the age recovered was 13.8 Ma, the closest to the age estimated here. Similarly, Mohammadi et al. [54] used four calibration schemes to calibrate the Lagomorpha node, adding a 52 Ma calibration point to those previously used by Melo-Ferreira et al. [16]. The youngest age obtained was 5.7 million years ago, while the oldest was 12.5 Ma. Ge et al. [23] reported an age of 13.4 Ma for Ochotonidae, which is also the middle Miocene.

There is a clear pattern in the rich fossil record of lagomorphs; after the divergence of the Leporidae and Ochotonidae, the two families diverged quickly, producing a few stem lineages during the Eocene and the Oligocene [23, 51, 52]. The Leporidae family appeared in the fossil record at approximately 37 Ma, during the upper Eocene period, followed by Ochotonidae at approximately 33 Ma, during the Oligocene period [51]. At the Oligocene and Miocene boundaries, distinct patterns emerged in each lineage. The Ochotonidae ancestor experienced a burst of diversification at the earliest Miocene, centred in Asia, followed by range expansions to Europe, North America and Africa [51, 56]. Between the middle and late Miocene, a second wave of ochotonids that included the extant genus *Ochotona* replaced several primitive genera in Asia and other continents [51]. In contrast, the Leporidae diversity remained relatively low throughout the Miocene but radiated profusely by the latest Miocene [23, 51, 52]. Leporids have also expanded their ancestral range from Asia to Europe, Africa and North America since the late Miocene [23, 51]. Therefore, the rich Lagomorpha fossil record clearly indicates that Leporidae is older than Ochotonidae, but that the major diversification of Ochotonidae predated that of Leporidae.

However, such a clear-cut pattern supported by a rich fossil record on multiple continents does not fit the current hypothesis for lagomorph diversification [23]. For instance, in that study, the diversification of Leporidae predated that of the extant *Ochotona* radiation. Although that study dated the *Ochotona* clade to ~13 Ma, a fossil of an undetermined *Ochotona* species from the Amuwusu Micromammals Site (Miocene of China) extends the age of the genus to ~16 Ma [57]. Since fossil age uncertainty and internal calibrations were not incorporated in the Ge et al. [23] time tree, such a recent age for *Ochotona* might have been an artefact. In our time tree, we incorporated multiple calibration points and fossil age uncertainties as probabilistic distributions. Our age for the *Ochotona* ancestor was estimated to be 15.4 Ma, which better fits the minimum age for the genus in the fossil record, especially because the *Ochotona* fossil from the Amuwusu site was not included in our calibration set. Last, in the Ge and coworkers time tree, the diversification of crown Leporidae was dated to approximately 18 Ma. Since stem representatives of Leporidae first appeared 37 Ma, that scenario requires a 19 million-year [Myr] gap in the fossil record for stem Leporidae, which is otherwise very rich on several continents. Our age for crown Leporidae was estimated at 25.4 Ma with a small

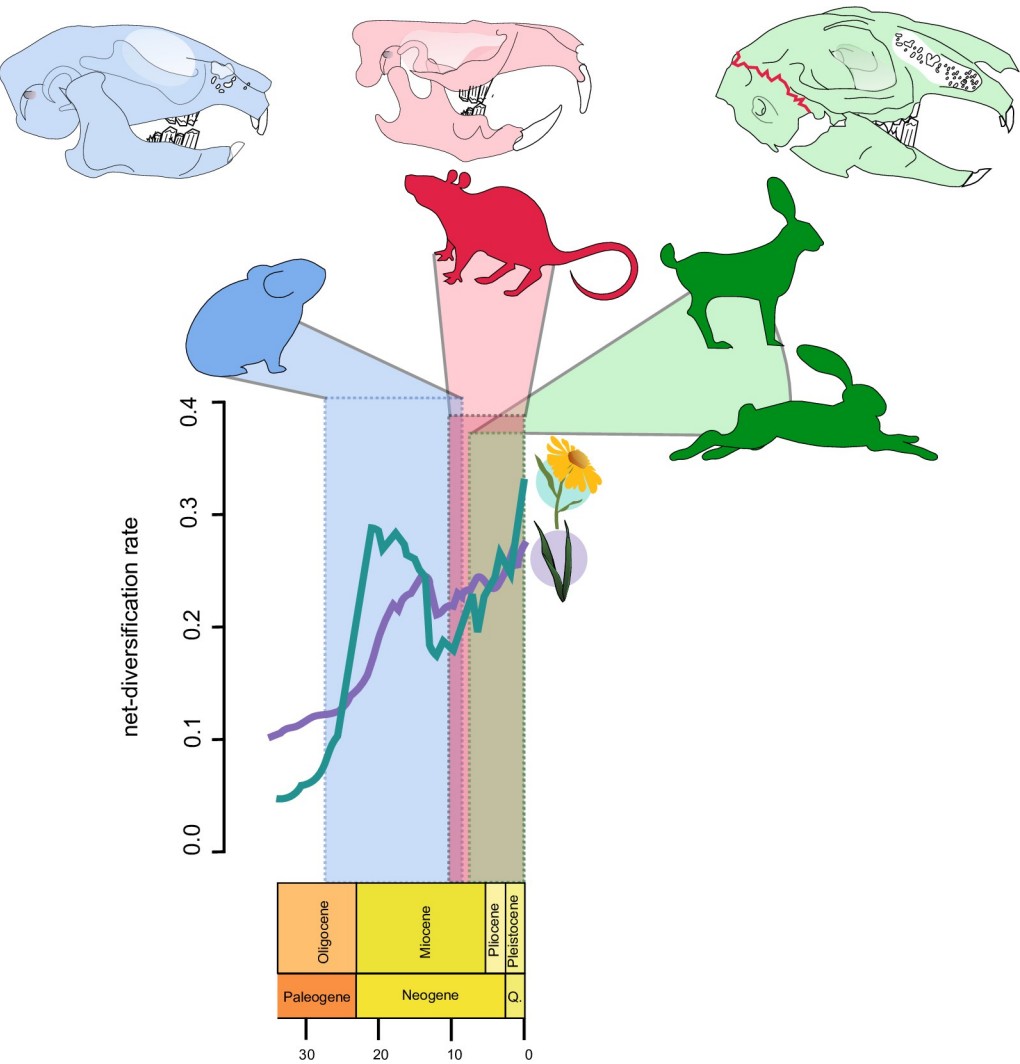

**Fig 3. Diversification scenario of Lagomorpha and grasslands during the late Cenozoic.**

confidence interval, thereby reducing the fossil gap to only 12 Myr (Fig 2). Therefore, our time tree better fits the fossil record for both crown clades of Lagomorpha.

In the upper Neogene, adaptive radiation started in the lineage of the genus *Lepus* and the lineage of *Sylvilagus*. The large genus *Lepus* originated at approximately 4.5 Ma during the Zanclean stage. Matthee et al. [20] reported the origin of this genus in Zanclean, with an age of 5.2 (±0.5) Ma. Ge et al. [23], on the other hand, recovered an age of approximately 8.6 million years ago, shifting the origin of *Lepus* to the Miocene epoch. The most recent estimated age for *Lepus* was that proposed by Cano-Sanchéz et al. [53], with the most recent common ancestor dating to 2.5 Ma (95% CI: 1–4). *Nesolagus* diverged from *Pronolagus* and *Poelagus* by 18.4 Ma, and *Pronolagus* and *Poelagus* diverged 15 Ma in the middle Miocene. The beginning of the diversification of the clade comprising *Pentalagus*, *Caprolagus* and *Sylvilagus* began at approximately 14.6 Ma, with *Sylvilagus* separating it from the other genera at 7.8 Ma. The divergence of this clade from *Sylvilagus* occurred at the beginning of the Tortonian Age, with *Sylvilagus* appearing at approximately 6.5 Ma in the Messinian Age in the Miocene, in contrast with the age of 5.3 (±0.8) Ma reported by Matthee et al. (2004). The latter is much closer to the age

recovered by Ge et al. (2013), with a *Sylvilagus* origin at 5.6 Ma. In addition, Cano-Sanchéz et al. [53] reported *Sylvilagus* as a paraphyletic taxon with respect to *Brachylagus*, and the age of this clade was ca. 1.5 Ma. The origin of the genus *Ochotona* was approximately 15.4 Ma, and its three subgenera had ages of 12.2 Ma (*Conothoa*), 11.8 Ma (*Pika*) and 6.1 Ma (*Ochotona*).

Among the seven time-resolved phylogenetic trees reconstructed in this study, there were some differences between the one that included all six fossils and the one in which one of the fossils was absent for the first lagomorph divergence (Table 2, Fig 1). This finding demonstrates the importance of including crucial calibration points. However, the congruence of most ages in the different calibration sets shows the robustness of the divergence time results of this part of the time tree.

The divergence time for crown Lagomorpha was consistent, approximately 58 million years ago, in almost all the sets, with only one exception. In the set without the Rodentia calibration point, there was a slight difference, with a younger age of 53 Ma. In the set without the calibration point of Lagomorpha itself, the oldest age is observed at approximately 62.5 Ma. Interestingly, the Leporidae family was consistent across all datasets, ranging from 23.7 to 27.4 Ma, and always recovered in the Chattian age. For Lagomorpha, when the Rodentia calibration was absent, the youngest age of 23.7 Ma was found. Ochotonidae showed the most stable age throughout the sets, at approximately 15.7 Ma, at the Langhian–Burdigalian boundary. The age of the *Lepus* genus was remarkably constant at 4.7 million years ago, always at the Zanclean age. The genus *Sylvilagus*, with a constant age of 6.7 Ma, differed in the set without the stem *Sylvilagus* calibration, with an age of 10.3 Ma.

## Mammal and plant radiation

The current macroevolutionary explanation for the distinct diversification patterns in the lagomorph fossil record is also tentative. In the late Miocene, low atmospheric $CO_2$ concentrations, seasonal fires and aridification fuelled grassland expansion worldwide, coinciding with the extinction of most ochotonids while leporids radiated [23]. That study suggested a direct link between C4 plant diversification, which predominates in the diet of leporids, and a decrease in the diversity of C3 plants, which are preferred by ochotonids (Fig 3). However, although C4 grasses are a major component of plant biomass in grasslands, recent studies have shown that many C3 grasses (e.g., Pooideae) also exhibited great diversity in the late Miocene [58]. Furthermore, other C3 plant families, such as daisies (Asteraceae), are often as diverse as (or more diverse than) grasses in grassland biomes [59]. Analyses of diversification rates and environmental variables have demonstrated that although grasses strongly differentiated during the late Miocene (~10 Ma), the same process was mirrored by the hyperdiverse Asteraceae [59]. Most importantly, surveys of food preference revealed Asteraceae as the dominant food preference in extant pikas [23, 60]. Additionally, the great diversity of Ochotonidae in the fossil record of the early Miocene closely matches the peak in diversity of Asteraceae within the same time frame ([59], Fig 3). Therefore, food preference alone does not explain the correlation between grassland expansion and the diversification–extinction patterns observed in Lagomorpha.

As most studies in Lagomorpha have focused on comparisons between its internal lineages, the macroevolutionary relationships between lagomorphs and other small mammals remain to be explored. The species-rich rodents (order Rodentia) are the leading model for small mammals due to their sheer morphological, ecological and geographic diversity [61–63]. Most importantly, they are the sister group of Lagomorpha that forms the clade Glires, and divergence time estimates based on nuclear and mitochondrial data suggest that the burst of adaptive radiation in rodents occurred in the last 10 Ma [64]. As Rodentia and Lagomorpha share

the characteristic of small body sizes and both have a pair of continually growing chisel-like incisors [61, 63], we suggest that such similarities in form and function are strong indicators of interspecific competition.

Therefore, to explain the diversity trends in Lagomorpha, we suggest a key biotic component that hinges on competition for resources with rodents. Fossils of stem Glires (e.g., Mimotonidae and Eurymylidae) indicate that small, rat-sized bodies and limb proportions are likely ancestral morphologies [61, 63, 65]. Palaeontological and molecular data agree that by the late Miocene, the Muroidea ancestor originated from the most diverse Rodentia families, Muridae and Cricetidae, which spread from Asia to other continents while consuming a variety of diets, from generalist omnivore diets (Muridae) to the strictly herbivorous diets consumed by Arvicolinae (Cricetidae) [62]. When this diversification pattern is compared with our chronological tree, we observed that the tempo and locale of the muroid radiation coincided with the drastic reduction in Ochotonidae genera in the fossil record (Fig 3).

Given that the two lineages share the ancestral Glires body plan and geographic ranges, we suggest that direct competition between members of the generalist and diverse Muroidea steered the mass extinction of the herbivorous Ochotonidae in the late Miocene. A recent study provided empirical evidence that rodent and pika populations interact enough to allow parasite spillover in North America [66]. An increase in rodent diversity has also been linked to extinction by competition with Multituberculata, a lineage closely related to the eutherian clade that disappeared when the major rodent lineages appeared in the fossil record worldwide [61, 63].

According to our hypothesis, while pikas competed for resources with muroid rodents of similar body sizes, the Leporidae ancestor would have escaped the turmoil of that competition by evolving a strikingly different body plan. Analyses of the complete fossil record of Glires in North America revealed a rise in Leporidae specialized in cursoriality (adapted to run) during the late Miocene [67]. Although Rodentia and Lagomorpha evolved burrowing habits and high crowned teeth (hypsodonty) suited to abrasive food such as that containing silica and dirt present in grasslands in the Miocene, only Leporidae showed increased cursorial specialization.

Leporidae species have overall larger body sizes than most Glires species [23, 68], and pikas and very few rodents exhibit true cursorial specializations due to their generally small size [68, 69]. Leporids, however, exhibit several unique adaptations to cursoriality, namely, a unique intracranial joint at the occipital region that alleviates spinal rebound during hopping [69] and extensive maxillary fenestration that provides mechanical resistance while reducing bone mass [70]. Although pikas exhibit some degree of maxillary fenestration, this characteristic became much more pronounced in the late Miocene Leporidae [69]. Finally, in the postcranial skeleton, the late Miocene Leporidae evolved elongated hindlimbs and metatarsals [68]. Therefore, the rapid diversification of the late Miocene Leporidae was marked by key innovations, such as powerful hindlimbs and cranium that withstand the mechanical pressures of high-speed locomotion. These are unique features of this lineage that, within our time tree, explained at least in part why leporids radiated and are more species-rich than their sister group.

The diversity trends in Lagomorpha might be related to a competition for resources with rodents. The decline in ochotonid diversity in the Late Miocene coincides with the radiation of the most diverse rodent families, Cricetidae and Muridae, which may have outcompeted the pikas particularly as the strictly herbivorous Arvocolinae murids diversified. Our findings are supported by the fossil evidence that indicate the ochotonid-rodent small body size as ancestral in Glires.

We suggest that Leporidae escaped the competition bottleneck and radiated by evolving a completely new morphospace—large bodies and an efficient cursorial locomotor system—

covering wide areas for foraging and outrunning predators more easily than rodents and pikas. Our results indicate that since the Pleistocene, rabbits and hares have radiated, especially those of the genera *Lepus* and *Sylvilagus*, which tended to remain taller than most rodents and pikas. Nevertheless, successful leporid body plans have been constrained by larger body sizes, as this would move them into competitive morphospace with ungulate-type herbivores [71].

## Conclusions

- Multiple markers and fossil priors recovered narrow confidence intervals and a consistently older age for the origin of Lagomorpha around 58 Ma.

- Varying the number of fossil calibrations (calibrations sets) had negligible effects on the ages of the major lagomorph lineages.

- Our age for the *Ochotona* ancestor fits the minimum age for the genus in the fossil record, especially because the oldest *Ochotona* fossil was not included in our calibration set.

- Our age for crown Leporidae was estimated at 25.4 Ma with a small confidence interval, thereby reducing the fossil gap to only 12 Myr.

- The Leporidae probably scaped competition with Ochotonidae and Muroid rodents when evolved a larger body and a body plan with a cursoriality specialization.

## Supporting information

**S1 Fig. Phylogenetic relationships of Lagomorpha.** Maximum likelihood tree generated using 79 terminals and 5 mitochondrial genes. Nodes with bootstrap values lower than 100 shown. In this topology, all species (including those that were considered rogue taxa) were kept.
(TIF)

**S1 Table. Species taxonomic assignments based on higher Taxonomy database.**
(DOCX)

**S2 Table. Access codes for sequences for five mitochondrial and 14 nuclear genes used in our analysis.**
(PDF)

## Acknowledgments

The authors are thankful for Prof. Lena Geise of the State University of Rio de Janeiro, in Brazil, for a helpful discussion regarding mammalian databases. This study was part of the Master's thesis of L.I. at the Biodiversity and Biologia Evolutiva Graduate Program at the Federal University of Rio de Janeiro.

## Author Contributions

**Conceptualization:** Leandro Iraçabal, Claudia Augusta de Moraes Russo.

**Data curation:** Leandro Iraçabal, Matheus R. Barbosa, Alexandre Pedro Selvatti.

**Formal analysis:** Leandro Iraçabal, Matheus R. Barbosa, Alexandre Pedro Selvatti.

**Funding acquisition:** Claudia Augusta de Moraes Russo.

**Investigation:** Leandro Iraçabal.

**Project administration:** Claudia Augusta de Moraes Russo.

**Writing – review & editing:** Alexandre Pedro Selvatti, Claudia Augusta de Moraes Russo.

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
