## [Decision Letter · Decision Letter 0]

11 Jun 2024

PONE-D-24-20437A molecular time-tree for LagomorphaPLOS ONE

Dear Dr. Russo,

Thank you for submitting your manuscript to PLOS ONE. After careful consideration, we feel that it has merit but does not fully meet PLOS ONE’s publication criteria as it currently stands. Therefore, we invite you to submit a revised version of the manuscript that addresses the points raised during the review process.

Please follow my recommendations in my own review.

We look forward to receiving your revised manuscript.

Kind regards,

Axel Janke

Academic Editor

PLOS ONE

Journal Requirements:

"This work was supported by the Coordenação de Aperfeiçoamento de Pessoal de Nível Superior – Education Ministry of Brazil (CAPES) – Finance Code 001 to student L.I. We are also thankful to the National Research and Technology Council (CNPq) process 310567/2018-1 to C.A.M.R. and a Master’s fellowship to M.R.B.C. and Rio de Janeiro State Research Funding Agency (FAPERJ) processes E-26/010.001887/2019, SEI-260003/001170/2020 (to Michelle Klautau), SEI-260003/012995/2021 to C.A.M.R. and E-26/203.840/2022 (277324) to A.P.S. This study was part of the Master’s thesis of L.I. at the Biodiversity and Biologia Evolutiva Graduate Program at the Federal University of Rio de Janeiro."

6. Please include a separate caption for each figure in your manuscript.

7. We note that Figure 2 in your submission contain copyrighted images. All PLOS content is published under the Creative Commons Attribution License (CC BY 4.0), which means that the manuscript, images, and Supporting Information files will be freely available online, and any third party is permitted to access, download, copy, distribute, and use these materials in any way, even commercially, with proper attribution. For more information, see our copyright guidelines: http://journals.plos.org/plosone/s/licenses-and-copyright.

Additional Editor Comments:

Dear Authors

I am sorry for the delay of the reviewing process, but it is hard to find find suitable and willing reviewer. To speed up the process, I will make my recommendation based on the one reviewer and my expertise.

REVIW OF THE EDITOR

The authors have assembled an impressive amount of sequence data from publicly available resources and have revisited the phylogeny of lagomorphs. This is a timely task, given the rapid accumulation of sequence data.

The analyses are straightforward, robust, and standard, resulting in a classic phylogenetic analysis based on the latest data. Since the focus was on a representative taxon sample, the number of loci (14) is limited, but still sufficient. Time calibration was performed using six fossil calibration points.

The results are nicely put into the context with the concurrent evolution of mammals and plants.

Lagomorpha is a large taxon with a long history of analysis and viewpoints, so my main recommendation/concern for improvement is to better point out differences in current understanding or controversial issues. This would help the non-specialist to better follow your findings.

Perhaps add a conclusion with bullet points for the five (or X) major findings.

Minor Issues:

- Give divergence times consistently with only three digits, because the estimates are not (cannot be) accurate to 10,000 years.

- Clean up the Suppl Fig., e.g. write some support values on the other side of the branch so that the numbers do not overlap.

- Enlarge the major clade to better see the divergence times, e.g. make a separate tree for Lepus. Or remove the outgroup Mus, Rattus, Acomys from the main figure (Fig. 1) to expand the branches for better readability and to show divergences from 30/25 Mya onwards.

These issues lead to a "minor revision" as they only improve the presentation of the results. However, I recommend to follow the recommendations (also of reviewer 1) for a better impact of the nice paper on the field.

Reviewers' comments:

Reviewer's Responses to Questions

**Comments to the Author**

1. Is the manuscript technically sound, and do the data support the conclusions?

Reviewer #1: Yes

2. Has the statistical analysis been performed appropriately and rigorously? 

Reviewer #1: Yes

3. Have the authors made all data underlying the findings in their manuscript fully available?

Reviewer #1: Yes

4. Is the manuscript presented in an intelligible fashion and written in standard English?

Reviewer #1: Yes

5. Review Comments to the Author

Reviewer #1: In the presented work, Leandro Iraçabal and coworkers have generated an ultrametric phylogenetic tree for Lagomorpha (hares, rabbits and pikas) using five mitochondrial and 14 nuclear loci. Besides constructing credible molecular phylogeny for the families and genera with high support values for branches, they reveal that the subgenus Ochotona is not monophyletic and would require revision. The tree was calibrated using available fossil evidence, and despite the usual caveats with such timings, seems like a plausible and realistic result.

The introduction is well written and well informed. The analysis used existing sequence data, without additional sampling and sequencing, which explains the targeting few loci in contrast to whole genome analysis. While the approach is perhaps somewhat outdated and does not provide novel sequence information, it is perfectly valid for the purpose and, thanks to the large amounts of available data, the analysis is also very comprehensive. The results are well presented and discussed. The further discussion of the Lagomorph evolution is very nice and highly useful as future reference. I have only a few minor suggestions make:

The title “A molecular time-tree” or the use of “time-tree” in the text appears somewhat imprecise use of terminology. Consider revising to e.g. chronophyletic tree, time-resolved phylogenetic tree or similar.

Introduction:

-Not only the rabbit, but also the European brown hare has been introduced to a number of locations, including Australia.

Results:

It would be highly useful to have more zoomed-in version of the tree in Fig 1 as supplementary figures to better see the estimated speciation times. This is especially interesting in genus Lepus, where the species frequently hybridize. It also better allows to see the effect of the last ice age on the species diversity. These figures could include the dates of the different stages of the Quartenary period.

Discussion:

-Please discuss more the effect of excluding different calibration points and how to evaluate the credibility of the differential results.

-Can the reticulation of especially the Lepus species through frequent hybridization affect the analyses, especially if this also has been occurring in the past?

-It seems likely that the fossil record be biased by the fact that Ochotonidae are much smaller than Leporidae? This could be pointed out.

-The plant codiversification is discussed in detail, but what has been the effect of the last ice age on Lepus radiation? There are a number of cold climate and snow adapted species on both sides of Beringia and the geography of the glacial refugia has probably equally affected the current diversity.

Overall, the manuscript presents a nice contribution towards the understanding of the Lagomorph evolution and origins of its species diversity.

6. PLOS authors have the option to publish the peer review history of their article (what does this mean?). If published, this will include your full peer review and any attached files.

Reviewer #1: No

---

## [Author Response · Author response to Decision Letter 0]

2 Jul 2024

R: Done. 

R: Done.

R: We could not find the Financial Disclosure section, but the Funding Information is now updated. 

4. Thank you for stating the following financial disclosure: "The author(s) received no specific funding for this work." At this time, please address the following queries:

a) Please clarify the sources of funding (financial or material support) for your study. List the grants or organizations that supported your study, including funding received from your institution. b) State what role the funders took in the study. If the funders had no role in your study, please state: “The funders had no role in study design, data collection and analysis, decision to publish, or preparation of the manuscript.”c) If any authors received a salary from any of your funders, please state which authors and which funders. d) If you did not receive any funding for this study, please state: “The authors received no specific funding for this work.” Please include your amended statements within your cover letter; we will change the online submission form on your behalf.

R: Done. We have included the statement “The funders had no role in study design, data collection and analysis, decision to publish, or preparation of the manuscript”. Some of the authors received fellowships from funding agencies. This is stated in the Funding Statement section. We could not find the Financial Disclosure section. 

5. Thank you for stating the following in the Acknowledgments Section of your manuscript: "This work was supported by the Coordenação de Aperfeiçoamento de Pessoal de Nível Superior – Education Ministry of Brazil (CAPES) – Finance Code 001 to student L.I. We are also thankful to the National Research and Technology Council (CNPq) process 310567/2018-1 to C.A.M.R. and a Master’s fellowship to M.R.B.C. and Rio de Janeiro State Research Funding Agency (FAPERJ) processes E-26/010.001887/2019, SEI-260003/001170/2020 (to Michelle Klautau), SEI-260003/012995/2021 to C.A.M.R. and E-26/203.840/2022 (277324) to A.P.S. This study was part of the Master’s thesis of L.I. at the Biodiversity and Biologia Evolutiva Graduate Program at the Federal University of Rio de Janeiro. "We note that you have provided funding information that is not currently declared in your Funding Statement. However, funding information should not appear in the Acknowledgments section or other areas of your manuscript. We will only publish funding information present in the Funding Statement section of the online submission form. Please remove any funding-related text from the manuscript and let us know how you would like to update your Funding Statement. Currently, your Funding Statement reads as follows: "The author(s) received no specific funding for this work." Please include your amended statements within your cover letter; we will change the online submission form on your behalf.

R: Done. We have removed the funding statement from the Acknowledgements section.

6. Please include a separate caption for each figure in your manuscript.

R: Done.

7. We note that Figure 2 in your submission contain copyrighted images. All PLOS content is published under the Creative Commons Attribution License (CC BY 4.0), which means that the manuscript, images, and Supporting Information files will be freely available online, and any third party is permitted to access, download, copy, distribute, and use these materials in any way, even commercially, with proper attribution. For more information, see our copyright guidelines: http://journals.plos.org/plosone/s/licenses-and-copyright. We require you to either (1) present written permission from the copyright holder to publish these figures specifically under the CC BY 4.0 license, or (2) remove the figures from your submission: a. You may seek permission from the original copyright holder of Figure 2 to publish the content specifically under the CC BY 4.0 license. 

 We recommend that you contact the original copyright holder with the Content Permission Form (http://journals.plos.org/plosone/s/file?id=7c09/content-permission-form.pdf) and the following text: “I request permission for the open-access journal PLOS ONE to publish XXX under the Creative Commons Attribution License (CCAL) CC BY 4.0 (http://creativecommons.org/licenses/by/4.0/). Please be aware that this license allows unrestricted use and distribution, even commercially, by third parties. Please reply and provide explicit written permission to publish XXX under a CC BY license and complete the attached form.” Please upload the completed Content Permission Form or other proof of granted permissions as an ""Other"" file with your submission. In the figure caption of the copyrighted figure, please include the following text: “Reprinted from [ref] under a CC BY license, with permission from [name of publisher], original copyright [original copyright year].” b. If you are unable to obtain permission from the original copyright holder to publish these figures under the CC BY 4.0 license or if the copyright holder’s requirements are incompatible with the CC BY 4.0 license, please either i) remove the figure or ii) supply a replacement figure that complies with the CC BY 4.0 license. Please check copyright information on all replacement figures and update the figure caption with source information. If applicable, please specify in the figure caption text when a figure is similar but not identical to the original image and is therefore for illustrative purposes only.

R: Done. Our original crania images were modified from the original copyrighted images, but they have been removed. The remaining images are of our own copyright. 

R: Done

R: Done. 

Additional Editor Comments: Dear Authors. I am sorry for the delay of the reviewing process, but it is hard to find find suitable and willing reviewer. To speed up the process, I will make my recommendation based on the one reviewer and my expertise. REVIEW OF THE EDITOR. The authors have assembled an impressive amount of sequence data from publicly available resources and have revisited the phylogeny of lagomorphs. This is a timely task, given the rapid accumulation of sequence data. The analyses are straightforward, robust, and standard, resulting in a classic phylogenetic analysis based on the latest data. Since the focus was on a representative taxon sample, the number of loci (14) is limited, but still sufficient. Time calibration was performed using six fossil calibration points. The results are nicely put into the context with the concurrent evolution of mammals and plants.

R: Thank you. 

Lagomorpha is a large taxon with a long history of analysis and viewpoints, so my main recommendation/concern for improvement is to better point out differences in current understanding or controversial issues. This would help the non-specialist to better follow your findings. 

R: Done. To build time-resolved phylogenetic trees from molecular datasets, geological ages are incorporated as calibration priors which are best provided by direct geological source such as the fossil record (primary calibrations). Previous studies used inadequate calibrations such as ages inferred from previous studies (secondary calibrations) and fossils that do not represent the earliest records of extant groups. Such suboptimal practices add much uncertainty to the estimates.

Perhaps add a conclusion with bullet points for the five (or X) major findings.

R: Done. We included bullet point conclusions. 

Minor Issues:

- Give divergence times consistently with only three digits, because the estimates are not (cannot be) accurate to 10,000 years.

R: Done. All time estimates are now with one decimal place. (Calibrations were kept with two decimal places.)

- Clean up the Suppl Fig., e.g. write some support values on the other side of the branch so that the numbers do not overlap.

R: Done. 

- Enlarge the major clade to better see the divergence times, e.g. make a separate tree for Lepus. Or remove the outgroup Mus, Rattus, Acomys from the main figure (Fig. 1) to expand the branches for better readability and to show divergences from 30/25 Mya onwards.

R: Done. The figure 2 is now a pruned tree for Leporidae.

These issues lead to a "minor revision" as they only improve the presentation of the results. However, I recommend to follow the recommendations (also of reviewer 1) for a better impact of the nice paper on the field.

Reviewers' comments. Reviewer's Responses to Questions

Comments to the Author

1. Is the manuscript technically sound, and do the data support the conclusions?

Reviewer #1: Yes

2. Has the statistical analysis been performed appropriately and rigorously? 

Reviewer #1: Yes

3. Have the authors made all data underlying the findings in their manuscript fully available?

Reviewer #1: Yes

4. Is the manuscript presented in an intelligible fashion and written in standard English?

Reviewer #1: Yes

5. Review Comments to the Author

Reviewer #1: 

In the presented work, Leandro Iraçabal and coworkers have generated an ultrametric phylogenetic tree for Lagomorpha (hares, rabbits and pikas) using five mitochondrial and 14 nuclear loci. Besides constructing credible molecular phylogeny for the families and genera with high support values for branches, they reveal that the subgenus Ochotona is not monophyletic and would require revision. The tree was calibrated using available fossil evidence, and despite the usual caveats with such timings, seems like a plausible and realistic result.

R: Thank you. 

The introduction is well written and well informed. The analysis used existing sequence data, without additional sampling and sequencing, which explains the targeting few loci in contrast to whole genome analysis. While the approach is perhaps somewhat outdated and does not provide novel sequence information, it is perfectly valid for the purpose and, thanks to the large amounts of available data, the analysis is also very comprehensive. The results are well presented and discussed. The further discussion of the Lagomorph evolution is very nice and highly useful as future reference. 

R: Thank you. 

I have only a few minor suggestions make: The title “A molecular time-tree” or the use of “time-tree” in the text appears somewhat imprecise use of terminology. Consider revising to e.g. chronophyletic tree, time-resolved phylogenetic tree or similar.

R: Done. We used time-resolved phylogenetic tree. 

Introduction:

-Not only the rabbit, but also the European brown hare has been introduced to a number of locations, including Australia.

R: Done. The sentence was rewritten. 

Results:

It would be highly useful to have more zoomed-in version of the tree in Fig 1 as supplementary figures to better see the estimated speciation times. This is especially interesting in genus Lepus, where the species frequently hybridize. It also better allows to see the effect of the last ice age on the species diversity. These figures could include the dates of the different stages of the Quartenary period.

R: Done. Figure 2 is the pruned Leporidae from Figure 1. 

Discussion:

-Please discuss more the effect of excluding different calibration points and how to evaluate the credibility of the differential results.

R: Done. We have now included a conclusion about the calibration scheme consistency. 

-Can the reticulation of especially the Lepus species through frequent hybridization affect the analyses, especially if this also has been occurring in the past?

R: High reticulation levels can certainly affect our analyses, since they were based on a phylogenetic tree which is not an appropriate way to depict the evolution of a reticulated group. However, our discussion and conclusions are probably unaffected as they focused on the macroevolutionary (above genera) patterns in Lagomorpha. 

-It seems likely that the fossil record be biased by the fact that Ochotonidae are much smaller than Leporidae? This could be pointed out.

R: Done. Given the same environment, larger animals are more likely to produce fossils simply because their body structures have more volume and therefore more chances to withstand decay until the fossilization process is concluded. However, if that were the case for Lagomorpha, we would expect that the fossil record of leporids would always be richer than the ochotonids’ in sites from multiple continents, that is not the case. Even though the oldest stem leporids are indeed older than the oldest stem ochotonids, the (fossil) diversity of leporids remained considerably low worldwide until the latest Miocene. On the other hand, there is an explosion in ochotonid fossil diversity worldwide from the Oligocene to the late Miocene. Furthermore, the large size of leporids only begin by the Late Miocene, reinforcing that there is no bias in the fossil record favoring this clade because their larger bodies is only a recent event in their 

---

## [Editor Report · Decision Letter 1]

4 Jul 2024

Molecular time estimates for the Lagomorpha diversification

PONE-D-24-20437R1

Dear Dr. Russo,

We’re pleased to inform you that your manuscript has been judged scientifically suitable for publication and will be formally accepted for publication once it meets all outstanding technical requirements.

Kind regards,

Axel Janke

Academic Editor

PLOS ONE
---

## [Editor Report · Acceptance letter]

28 Aug 2024

PONE-D-24-20437R1 

PLOS ONE

Dear Dr. Russo, 

I'm pleased to inform you that your manuscript has been deemed suitable for publication in PLOS ONE. Congratulations! Your manuscript is now being handed over to our production team.

Kind regards, 

on behalf of

Dr. Axel Janke 

Academic Editor

PLOS ONE